# Community Involvement in Onchocerciasis Post-elimination Surveillance in Bududa District, Eastern Uganda: A cross-sectional study

Annet Tabitha Khainza[1,2]*, David Soita[1], David Okia[1], Francis Okello[1], Joseph KB Matovu[1], Yovani Lubaale[1], Edson Byamukama[2], Ambrose Okibure[1], Jimmy Patrick Alunyo[1], Ritah Nantale[1], Benon Wanume[1], David Ogutu[3], David Mukunya[1], Peter Olupot-Olupot[1,4]

1 Department of Community and Public Health, Busitema University, Mbale, Uganda, 2 The Carter Center, Kampala, Uganda, 3 Ministry of Health, Kampala, Uganda, 4 Department of Research, Mbale Clinical Research Institute, Mbale, Uganda

* annetkhainza84@gmail.com

**Data Availability Statement:** All relevant data are within the paper and/or its Supporting Information files.

## Abstract

### Background

Globally, there are an estimated 20.9 million cases of onchocerciasis, with Africa bearing the greatest burden. The World Health Organization (WHO) has targeted the disease for elimination by 2030. As of August 2023, there were 15 foci in 37/48 (76%) districts and one city in Uganda that had reached the elimination phase. However, there is a paucity of data on community involvement in post-elimination surveillance (PES) activities. The communities in the post-elimination phase are expected to maintain surveillance, provide health education, refer cases for treatment, and participate in surveillance. However, it is not clear whether this is being done. In this study, we assessed the feasibility of community involvement in post-elimination surveillance activities in Bududa District, Eastern Uganda, to draw key generalisable lessons for similar settings.

### Methods

This was a cross-sectional study employing rigorous mixed methods of data collection. We used a semi-structured questionnaire to collect quantitative data on randomly sampled study participants in two sub-countries in the district. Community involvement in post-elimination surveillance (PES) was our dependent variable, measured using Yes or No questions, and our independent variables were measured on different scales. Computations of proportions and associations were done using Stata 15 software. Conversely, qualitative data were collected via focus group discussions (FGDs) for community participants and key informant interviews (KIIs) for local leaders. For the qualitative component, we had 2 FGDs, each consisting of 8 gender-balanced participants per group and 8 KIIs. Qualitative data analyses were done using a robust thematic framework approach, ensuring the reliability and validity of our findings.

**Funding:** The author(s) received no specific funding for this work.

**Competing interests:** The authors have declared that no competing interests exist.

## Results

A total of 422 participants with a mean age of 51.4 years (SD = 15.8) participated in the study. Community involvement in post-elimination surveillance was low (14%). Factors associated with involvements were district support [Adjusted odd ratio AOR 14, 95 CI = (2.5, 81.7)], seeing black flies in the environment in a week preceding the survey [AOR 8, 95% CI = (1.5, 42.5)], in one month [AOR 3.8, 95% CI = (1.1, 13.2)], and being a community volunteer in the Ivermectin treatment program [AOR 4.3, 95% CI = (1.03, 17.9)]. Lack of funding, poor motivation, poor program sustainability planning, and a lack of drugs at health facilities were key challenges affecting community involvement in post-elimination surveillance.

## Conclusion

Community involvement in onchocerciasis post-elimination surveillance activities in Bududa District in Eastern Uganda was low but could be improved by increased district support, funding, community motivation and sensitisation.

### Author summary

This study addresses the crucial issue of onchocerciasis, a neglected tropical disease, with a focus on post-elimination surveillance in Bududa District, Eastern Uganda. While global efforts target onchocerciasis elimination by 2030, this research assesses community involvement in post-elimination surveillance, uncovering a low participation rate of 14% in Bududa.

Factors influencing involvement include district support, black fly sightings, and community volunteer experience. Challenges, such as funding shortages and program sustainability issues, hinder community engagement. Additionally, the study emphasizes the need for increased district support, funding, and community sensitization to improve participation. The findings provide valuable insights for similar settings.

With a total of 422 participants, the study explores both quantitative and qualitative data, offering a comprehensive perspective. Ultimately, the research contributes essential knowledge to enhance community participation and strengthen post-elimination surveillance strategies in the fight against onchocerciasis in Uganda.

## Background

Onchocerciasis, also known as river blindness, is a neglected tropical disease (NTD) primarily caused by a parasitic worm, *onchocerca volvulus* (O. volvulus) [1,2].This disease spreads through bites by infected black flies of the *Simulium genus*, which thrive in the vicinity of fast-flowing streams and rivers[3] These flies commonly inhabit remote rural tropical areas. The infection, if untreated, can lead to visual impairment or blindness [1]. In addition, it often causes skin disease, including intense itching, rashes, or nodules under the skin [1]. Additionally, some studies have also linked onchocerciasis to neurohormonal epilepsy, like nodding disease syndrome in northern Uganda [4].

In 2017, the Global Burden of Disease study estimated a worldwide total of approximately 20.9 million prevalent O. volvulus infections. Among these cases, 14.6 million individuals experienced skin diseases, and 1.15 million had vision loss [5]. The African continent bears

almost the entire global burden of onchocerciasis, accounting for 99% of the global burden of onchocerciasis [6]. Notable progress has been made in the Americas. In 13 distinct areas throughout Brazil, Colombia, Ecuador, Guatemala, Mexico, and Venezuela, approximately 500,000 individuals faced the risk of onchocerciasis infection.[6]. In 2012, the effective elimination of the disease in the Americas was accomplished through bi-annual and sometimes quarterly treatment with ivermectin for at-risk populations [6], which served as a successful example that was replicated in Africa. The African continent adopted a comparable strategy of once or twice a year treatment with ivermectin to treat affected communities. Still, it complemented it with a vector control/elimination initiative compared to the Americas. However, it is worth noticing that MDA and vector control were adopted only in countries under the onchocerciasis control program, especially in West Africa. This initiative entailed treating breeding sites along rivers with organophosphate insecticide *temephos* (Abate EC200) [7]. However, despite temephos' effectiveness in reducing vector populations and lowering the risk of disease transmission, its wide use raises environmental concerns. These dual strategies played an instrumental role in the elimination of onchocerciasis in some areas of Africa [7,8]. At the start of Uganda's elimination efforts in 2007, a population of 4.9 million people was at risk. The area included 16 foci, excluding the Victoria Nile focus, located approximately 80 km from the source of the River Nile near Jinja town. This area extended to a point where the river slowed down before reaching Lake Kyoga [9]. The population in this area faced the risk of onchocerciasis. However, as of August 2023, the number of people at risk had dwindled to 697,032 people in the country. This achievement emerged from a collaborative endeavour involving public and private organisations, local communities, donor groups, Inter-governmental bodies, academic institutions, non-profit organisations, and pharmaceutical companies. The goal of eliminating onchocerciasis in the 12 African countries where the disease is endemic is set for 2030 [8]. To achieve this objective, the World Health Organization (WHO) has recommended that national governments implement post-elimination surveillance activities in areas where interruption of onchocerciasis has been attained, extending this effort to cover the entire country before declaring it free of onchocerciasis. The process leading to onchocerciasis elimination, as recommended by WHO, begins with mapping the disease and implementation of mass drug administration (MDA) using ivermectin (intervention phase). This is followed by continuous monitoring and evaluation. Surveys to halt MDA which involve assessment of the black fly populations in the community through entomological methods and evaluation of children epidemiologically using serological test [8].Upon successful implementation of this phase, community-directed treatment with ivermectin (CDTI) is then discontinued, and the onchocerciasis-endemic area will enter post-treatment surveillance (Phase 2), which will run for three to five years. During this phase, entomological surveillance is then conducted to confirm the interruption of transmission. After this, the program will then move to post-elimination surveillance (Phase three), which signifies the permanent halting of treatment. Occasional surveys during Phase 3 will now be conducted to ensure that there is no resurgence or re-establishment of infections before officially declaring regional elimination [10]. By the close of 2017, three countries -Republics of Venezuela, Uganda, and Sudan had already ceased mass drug administration (MDA) and completed three years of post-treatment surveillance in at least one transmission area [3]. The positive outcomes observed in these affected regions can be partially attributed to strong collaborations among diverse stakeholders, including international donors, the WHO, governments, NGOs and communities [3], as there could have also been other drivers of this success, such as less competent vectors, as witnessed in the Americas.

In 2007, Uganda initiated a nationwide onchocerciasis elimination policy centered around semiannual treatment and vector control/elimination, aiming at eliminating onchocerciasis

[11]. The adoption of this bi-annual treatment with ivermectin in Uganda was based on the World Health Organization (WHO) guidelines [8], which typically advocated for annual or biannual treatment depending on the endemicity and transmission dynamics of onchocerciasis in a given area. For instance, the whole district of Bududa was endemic at the time. Since the adoption of the policy, 37(76%) districts out of 48 districts and one city have attained onchocerciasis elimination status and are under post-elimination surveillance (PES) [12]. As of 2023, the country still had seven districts in post-treatment surveillance (PTS), and interventions are ongoing in 5 border districts where transmission interruption is suspected. Throughout these efforts, there has been uncertainty regarding the role and extent of communities in post-elimination surveillance (PES).

Furthermore, it was not known whether the capacity (maintaining village-based surveillance, creating community awareness about the disease and involvement in the implementation of the intervention and periodic surveys to monitor recrudescence)created during the post-treatment surveillance (PTS) period was sufficient to enable the communities to implement sound and vibrant post-elimination surveillance activities aimed at preventing possible recrudescence in the districts where the disease has been eliminated. The national onchocerciasis elimination guideline emphasises the importance of strengthening national surveillance at all levels to ensure onchocerciasis detections and appropriate action. This surveillance was supposed to be created with the assistance of village health teams (VHTs), parish community supervisors and community members. This study, therefore, aimed at assessing the level of community involvement in PES activities within Bududa district, identifying the challenges encountered, and proposing recommendations to strengthen post-elimination in other districts that eliminated the disease.

## Methods

### Ethics statement

Ethical approval was obtained from the Research and Ethics Committee of Busitema University, approval number BU-2022-043. We sought administrative clearance from the DHO, CAO, LC5 chairpersons and local leaders in Bududa district. Written informed consent was obtained from all participants in the study before collecting data. Participation in the study was voluntary; participants had a right to withdraw from the study at any time, identifiable information such as participant's names was not collected, and maximum confidentiality of the information collected was ensured to all participants throughout the study.

### Study design

This was a cross-sectional study that utilised both quantitative and qualitative data collection methods. A semi-structured tool was developed to collect quantitative data, and a key informant guide and focus group discussion guides were used to collect qualitative data.

### Study setting

The study was conducted in Bududa district, eastern Uganda. Bududa is one of the districts that form Mount Elgon onchocerciasis transmission area. Mt. Elgon focus is among the first foci that eliminated onchocerciasis in 2016 and has been under PES for over six years. The Mt. Elgon focuses on Uganda, covering an estimated area of 1,500 km2, with a mean onchocerciasis mean prevalence of 80% [2], and classified as hyperendemic. The intervention started with vector control using dichlorodiphenyltrichloroethane (DDT) because of the species *S. Neavei* around the year 1957. This was followed by the introduction of annual mass treatment with

ivermectin, which was changed to two times per year, along with a vector control elimination strategy that was reintroduced again in 2007 after its success in the Victoria focus. Bududa had a population of about 223,734 people who were estimated to have been at risk of onchocerciasis [12]

The district is bordered by Sironko district to the north, Kenya to the east, Manafwa district to the south and Mbale district to the west. The district has one general hospital, 12 Health Centre III, and four Health Centre II, with Bududa Hospital as the main hospital. The main river responsible for breeding the vector is Namatyale and its tributaries. At the time of the elimination of onchocerciasis, Bududa had seven sub-counties of Bududa, Bushika, Bukigai, Buluckeche, Bukibokolo, Bubita and Bukalasi, 42 parishes and 412 communities with a total projected population of 223,734 [12]. Bududa district is predominantly occupied by the Bagisu, who are mainly peasant farmers.

## Quantitative component

**Study population.**   These were individuals who lived in Bududa while onchocerciasis was still endemic in the communities, and the communities themselves were involved in its elimination activities. We included individuals who were involved in onchocerciasis community activities until PES was initiated and lived in the community at the time of the intervention period. These included former ivermectin users, community supervisors and community drug distributors (CDDs) who had lived in the district during the entire treatment period. We excluded visitors, migrants, and individuals who had lived in the area when the onchocerciasis was interrupted because they had no knowledge and experience of the onchocerciasis disease.

**Sample size estimation.**   The sample size for this study was determined using Kish Leslie formula (1965). We assumed 50.0% as the proportion of community involvement since no previous study investigated the proportion of community involvement in PES activities before in Bududa district, with a 5% precision of the estimate and 95% confidence interval level. This gave us a sample size of 384 participants. Considering a non-response rate of 10%, the estimated sample size was 422 participants.

**Sampling procedure.**   At the time of elimination, Bududa had seven sub-counties, 42 parishes and 412 villages. Therefore, to identify the sample population, we conducted a multistage sampling, which involved purposive sampling at the sub-county level and simple random sampling at the parish and village levels. We purposively selected two sub-counties of Bukigai and Bushika due to their proximity to the formerly black fly breeding river of Namatyale, and they were all endemic at baseline. A total of seven parishes and 25 villages were obtained using simple random sampling. To generate the lists of households at the village level, we approached local leaders who, with the help of Village Health Teams (VHTs), had previously worked on an onchocerciasis program and lived in the community during the entire treatment period when onchocerciasis was still endemic enlisted qualifying participants. From this list, simple random sampling was applied to select the study participants using random numbers. A total of 422 participants were sampled based on the population proportion in each sampled village.

**Study variables.**   The dependent variable was community involvement in post-elimination surveillance, which was measured quantitatively as a binary outcome. For those who were involved in onchocerciasis, PES activities were coded Yes, while "No" was for those who were not involved.

Independent variables were social demographic characteristics (age, marital status, residence, level of education, income level, etc.) of the participants, knowledge, and awareness about onchocerciasis post elimination surveillance activities.

**Data collection tool and procedures.**   The quantitative data was collected using an interviewer-administered semi-structured questionnaire, which was entered into the electronic

data collection tool Kobo Toolbox (Cambridge, Massachusetts, USA). The questionnaire included questions on the socio-demographic factors (age, marital status, tribe, educational level, income levels, occupation, and religion), knowledge and awareness about onchocerciasis post-elimination surveillance activities and implementation of onchocerciasis post-elimination surveillance activities. The tool was piloted in Bufumbo-sub-county before actual data collection.

The data collection process was assisted by six trained research assistants and two field supervisors who were trained for three days on the study procedures, ethical issues, and how to use the data collection tool and app. After completing the training, they were deployed to the field, assisted by the Local Council (LC1) chairperson of the village and former community supervisors to locate the households selected to participate in the study. To ensure the quality of data that was collected, daily data validation checks in the Kobo Toolbox and troubleshooting were done. Data was backed up in an external drive created in the Principal Investigator's computer, which was protected with a password to ensure confidentiality. Other backup data was created on flash discs that have limited access.

**Data analysis.** To compute descriptive statistics, categorical variables were summarized using frequencies and percentages, while continuous variables were summarized using mean (standard deviation). Community involvement was a dichotomous variable and was categorized as Yes and No. Community involvement was calculated as a proportion to determine the level of involvement. At the bivariate level, the Chi-square test was applied to calculate proportions and, in some cases, Fisher exact, where the values in our tables were less than five. The number of participants who participated in post-elimination surveillance activities was divided by the total number of participants enrolled in this study. Multivariable logistic regression was applied to determine the factors associated with community involvement in PES activities while controlling for confounders. We opted for logistic regression due to the binary nature of our primary outcome—community involvement in post-elimination—measured on a Yes/No scale. Factors with a significance level (p-value) below 0.05 in the bivariable analysis were incorporated into the multivariable analysis, employing forward selection criteria. Calculations for adjusted odds ratios (AOR), 95% confidence intervals, and p-values were performed, considering statistical significance at a threshold of p-value<0.05.

## Qualitative component

**Study population.** The participants selected as key informants were individuals who took part in the onchocerciasis elimination program during the time the interventions were ongoing through the post-treatment surveillance period either as sub-county onchocerciasis focal persons/health assistants or parish supervisors (PS). The participants for focus group discussions (FGDs) and key informant interviews (KIIs) were obtained across the selected communities from six out of seven parishes within the study area. The seventh parish was not considered because of difficulties in accessing it. We included participants who previously participated in onchocerciasis community activities either as beneficiaries, distributors, or supervisors. The participant must have lived in the areas as of then and still resided and participated in onchocerciasis community activities until PES was initiated. They must have also been mentally stable and consented to participate in the study. We excluded individuals who were non-residents and did not previously participate in the onchocerciasis program. We also excluded those who did not consent, mentally handicapped, and immigrants.

**Sample size determination.** The sample size was determined by the principle of data saturation, and we conducted two focus group discussions (FGDs) and 8key informant guides (KIIs) Table 1.

**Table 1. The sample size for the qualitative interviews.**

| Type of interview | Sub-county 1 | Sub-county 2 | Total | Categories of people |
|---|---|---|---|---|
| Focus Group Discussion | 8 (females) | 8 (males) | 16 | Community members |
| Key Informants | 3 Parish Supervisors (2 females and one male) | 3 Parish Supervisors (1 female and two males) | 6 | Parish Supervisors |
| | Health Workers (1 male) | Health Workers (1 female) | 2 | Health Workers |

**Data collection procedures and methods.** Data were obtained using FGD/KII interview guides. Information regarding various aspects of challenges, like issues of sensitization, motivation and socioeconomic factors, and human and material resources, were captured. The purpose of FGDs was to explore the group perspectives on the challenges of post-elimination surveillance activities (PES) in the district. While KIIs explored detailed knowledge regarding the challenges of implementing PES activities. The topic guides were reviewed by the research team prior to the interviews and were translated into the native language. The interview guides were piloted in Bufumbo Sub-County to ensure clarity. All the interviews were audiotaped for reference and to ensure that participants' views and opinions were not misreported and misinterpreted during transcription. The interviews and FGDs that were conducted in the local language were transcribed and coded thematically. Regular meetings were held to harmonize the coding and to discuss any disagreements. This helped to improve the credibility of our results and increased its dependability.

**Data analysis.** Thematic analysis, following the well-established six-step framework outlined by Braun and Clarke (2006), served as the primary analytical approach. The initial phase involved immersing oneself in the data, including transcriptions, reading, and repeated listening to the audio recordings. Preliminary codes were generated during this phase, capturing interesting and meaningful features of the data. These codes provided valuable insights into the contextual tones of the conversations, laying the foundation for subsequent analysis.

The search for themes commenced the interpretive analysis process, whereby relevant data extracts were carefully sorted and organized into main themes. A comprehensive review of the identified themes ensued, wherein initial themes were carefully evaluated for refinement, combination, separation, or potential exclusion. This review process was conducted in two distinct phases. Firstly, themes were cross-checked against the coded extracts at the individual interview level (Phase 1), ensuring coherence within the context of each interview. Subsequently, the themes were reviewed again in relation to the overall dataset (Phase 2), thereby ensuring a comprehensive and robust analysis.

The subsequent step involved the definitive definition and naming of the themes. The researcher precisely refined the themes and identified potential subthemes within the data, accompanied by clear working definitions. These definitions concisely captured the essence of each theme, aligning them with the research question and relevant literature. Furthermore, the presentation of the findings included the identified themes supported by pertinent quotations, thereby enhancing the credibility and transparency of the analysis.

## Results

### Participant characteristics

Of the 422 people who participated in this study, more than half of 216/422(51.2%) were females, and the rest were males. Approximately half of the participants, 212/422 (50.2%), belonged to the age group of 35–59, and the majority 366/422(86.7%) were married. In terms of literacy, the majority had attended school (84.8%), with primary education (64.9%) being the highest level completed Table 2.

**Table 2. Socio-demographic characteristics of the participants.**

| Variable | All sample n = 422 | Percentage (%) |
|---|---|---|
| Sex of Respondent | | |
| Male | 206 | 48.8 |
| Female | 216 | 51.2 |
| Age of respondent (complete years) | | |
| 20–34 | 71 | 16.8 |
| 35–59 | 212 | 50.2 |
| 60+ | 139 | 32.9 |
| Marital status of the respondent | | |
| Married | 366 | 86.7 |
| Separated | 13 | 3.1 |
| Widowed | 37 | 8.8 |
| Single | 6 | 1.4 |
| Religious Affiliation | | |
| Born again/Pentecostal | 50 | 11.8 |
| Catholic | 161 | 38.2 |
| Protestant | 173 | 41.0 |
| Muslim | 22 | 5.2 |
| Others | 16 | 3.8 |
| Ever attended school (Yes) | 358 | 84.8 |
| Highest education completed | | |
| No formal education | 67 | 15.8 |
| Primary | 274 | 64.9 |
| Secondary | 70 | 16.6 |
| Technical/vocational Cert | 4 | 0.9 |
| University/tertiary | 11 | 4.5 |

## PES community involvement

Community involvement in onchocerciasis PES activity was very low at only 14% (58/422), and the majority 86% (364/422) were not participating in PES **Fig 1**.

## Knowledge and awareness about onchocerciasis disease

Out of 422 participants, 339/442 (76.7%) reported to have heard about the onchocerciasis disease. More than half 223/442 (66%) also knew what causes the disease. Regarding the availability of the black fly in the community, 44/422 (13.7%) reported that the black fly that causes onchocerciasis was still common in their area. A majority of the participants, 288/422 (89.4%), reported seeing the black flies years back. Only 21/422 (6.5%) had seen them within a month ago, while 13/422 (4%) reported seeing them within a week.

More than half of the participants 237/422 (73.6%) knew the breeding ground for the black flies. A few of the participants, 28/422 (6.6%), had experienced signs and symptoms of onchocerciasis Table 3.

## Knowledge of Implementation of onchocerciasis PES Activities

The results of the study revealed that a majority of the participants (44.3%) identified skin rashes as the primary indicator of suspected river blindness, followed by extreme itching (38%) and itching of the eyes (29%). When faced with a suspected case of onchocerciasis, the

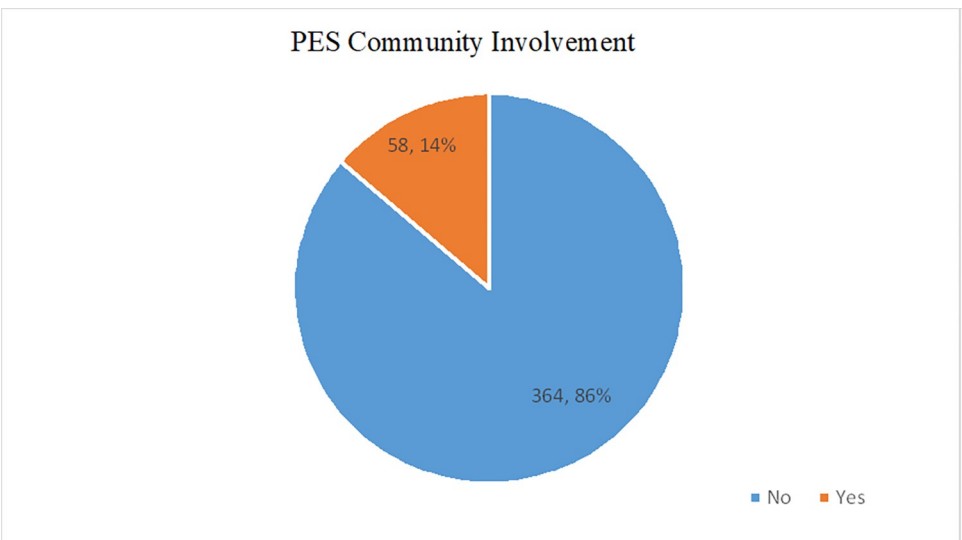

**Fig 1. A Pie Chart showing community involvement in PES.** Yes: Proportion of community members who were involved in PES activities(14%), No: Proportion of community members who were not involved in PES activities (86%).

preferred course of action among participants was to refer the patient to a nearby health facility (28%), followed by informing the VHT (23.4%) and local leaders (8%). It is worth noting that only a small proportion of participants (17%) still perceived river blindness as a prevailing issue within their community Table 4.

## Implementation of onchocerciasis PES Activities

Out of the total sample size of 422 participants, 21(5.9%) reported seeing a person with onchocerciasis signs months back, and the majority, 181 (51.3%), reported seeing an individual

**Table 3. Knowledge and awareness about onchocerciasis disease.**

| Variable | All sample n = 422(%) | PES community involvement | | P-value |
|---|---|---|---|---|
| | | No n = 369(%) | Yes n = 58(%) | |
| Ever heard about river blindness | | | | 0.392 |
| (Yes) | 339(80.3) | 290(79.7) | 49(84.5) | |
| No | 83(19.7) | 79(21.3) | 9(15.5) | |
| Knows what causes river blindness. | | | | 0.004 |
| (Yes) | 223(65.8) | 182(62.8) | 41(83.7) | |
| No | 199(34.2) | 187(37.2) | 17(16.3) | |
| Heard of a blackfly | | | | 0.362 |
| (Yes) | 322(76.3) | 275(75.5) | 47(81.0) | |
| No | 100(23.7) | 94(24.5) | 11(19.0) | |
| Black flies are still common in the area (Yes) | 44(13.7) | 26(9.5) | 18(38.3) | 0.001 |
| Ever seen the Black flies | | | | 0.001 |
| Days back | 13(4.0) | 6(2.2) | 7(14.9) | |
| Months back | 21(6.5) | 15(5.5) | 6(12.8) | |
| Years back | 288(89.4) | 254(92.4) | 34(72.3) | |
| Knows the breeding ground of the black flies (Yes) | 237(73.6) | 191(69.5) | 46(97.9) | 0.001 |
| Have you ever experienced any of the onchocerciasis symptoms (Yes) | 28(6.6) | 20(5.5) | 8(13.8) | 0.018 |

P-values based on Chi-square test (Chi2)

**Table 4. Knowledge of Implementation of onchocerciasis PES Activities.**

| Variable | All sample n = 422(%) | PES community involvement | | P-value |
|---|---|---|---|---|
| | | No n = 369(%) | Yes n = 58(%) | |
| Ways of identifying a person suspected of river blindness | | | | 0.001 |
| Skin rashes | 187(44.3) | 148(40.1) | 39(67.2) | |
| Extreme itching | 59(14.0) | 32(8.8) | 27(46.6) | |
| Nodules under the skin | 56(13.3) | 41(11.3) | 15(25.9) | |
| Extreme itching | 159(37.7) | 125(34.3) | 34(58.6) | |
| Bumps under the skin | 78(18.5) | 55(15.1) | 23(39.7) | |
| Loss of skin elasticity, which can make skin appear thin and brittle | 50(11.9) | 30(8.2) | 20(34.5) | |
| Itching of the eyes | 122(28.9) | 97(26.7) | 25(43.1) | |
| changes to skin pigmentation /leopard skin | 72(17.1) | 53(14.6) | 19(32.8) | |
| Enlarged groin | 50(11.9) | 29(8.0) | 21(36.2) | |
| Cataracts | 29(6.9) | 12(3.3) | 17(29.3) | |
| Light sensitivity | 44(10.4) | 22(6.0) | 22(37.9) | |
| What is done in the case of a person with signs of river blindness | | | | 0.001 |
| Inform Local leaders | 31(7.7) | 25(7.3) | 6(10.5) | |
| Refer the person to the nearest health facility | 112(27.9) | 87(25.3) | 25(43.9) | |
| Inform VHT | 94(23.4) | 79(23.0) | 15(26.3) | |
| Refer the person to the VHT | 13(3.2) | 8(2.3) | 5(8.8) | |
| Inform the Health worker | 11(2.7) | 6(1.7) | 5(8.8) | |
| Do Nothing | 9(2.2) | 7(2.0) | 2(3.5) | |
| Others | 3(0.8) | 3(0.9) | 0(0.0) | |
| Health workers in the community support the suspected river blindness (Yes) | 241(57.1) | 207(56.9) | 34(58.6) | 0.802 |
| Form of support offered by the health worker in the community | | | | 0.017 |
| Refer onchocerciasis suspected cases to the laboratory for testing | 46(19.1) | 36(17.4) | 10(29.4) | |
| Provide first treatment | 36(14.9) | 28(13.5) | 8(23.5) | |
| Treatment | 121(50.2) | 109(52.7) | 12(35.3) | |
| Counseling | 4(1.7) | 2(1.0) | 2(5.9) | |
| Don't Know | 34(14.1) | 32(15.5) | 2(5.9) | |
| Think that river blindness is still a problem within the community | 70(16.6) | 40(11.0) | 30(51.7) | 0.001 |

displaying signs and symptoms of onchocerciasis years back. In contrast, 151 (42.8%) were not sure. In terms of information dissemination, most of the participants 361/422 (85.6%) reported receiving disease-related information through VHTs, while 126/422 (29.9%) obtained information from radio programs. Although many participants demonstrated awareness of malaria as the primary health program conducted in the community (293 out of 422, or 88%), only a small proportion (18 out of 422, or 5%) reported being aware of River blindness, which was integrated into the implementation of malaria programs. Regarding information sources, the primary modes of communication for health program information within the community were VHTs (361 out of 442, or 85.6%), followed by radio communication (126 out of 422, or 29.9%) and community routine dialogues (73 out of 422, or 17.3%).

The study identified several challenges in conducting onchocerciasis health education, including loss of interest (45%) and lack of time (30%). Participants who expressed disinterest or lacked sufficient time were less likely to engage in onchocerciasis health education Table 5

**Table 5.  Implementation of onchocerciasis PES Activities.**

| Variable | All sample n = 422(%) | PES community involvement | | P-value |
|---|---|---|---|---|
| | | No n = 369 (%) | Yes n = 58 (%) | |
| Last time, one saw people with signs and symptoms. | | | | 0.020 |
| Months backs | 21(5.9) | 13(4.4) | 8(14.3) | |
| Years back | 181(51.3) | 159(53.5) | 22(39.3) | |
| Not sure | 151(42.8) | 125(42.1) | 26(46.4) | |
| Roles played by individuals before 2016 | | | | |
| Participated as Sub- County onchocerciasis focal person during the most recent intervention phase (*Yes*) | 3(0.7) | 1(0.3) | 2(3.5) | 0.001 |
| Participated as Community supervisor during most recent intervention phase (*Yes*) | 9(2.1) | 3(0.8) | 6(10.3) | 0.001 |
| Participated as Drug distributors during the intervention phase (*Yes*) | 49(11.6) | 28(7.7) | 21(36.2) | 0.001 |
| Participated as a Community volunteer during the intervention phase (*Yes*) | 13(3) | 4(1.1) | 9(15.5) | 0.001 |
| Never Participated in any activity during the most recent intervention phase (*Yes*) | 321(76.1) | 293(80.5) | 28(48.3) | 0.001 |
| Currently playing any role even after 2016 | 28(6.6) | 8(2.2) | 20(34.5) | 0.001 |
| Role played | | | | |
| Mobilizer | 8(28.6) | 2(25) | 6(30.0) | 0.958 |
| Referral | 5(17.9) | 1(12.5) | 4(20.0) | |
| Health education | 6(21.4) | 2(25.0) | 4(20.0) | |
| Others | 5(17.9) | 2(25.0) | 3(15.0) | |
| Mode of receiving information in the community about health programs | | | | 0.001 |
| Community meetings | 73(17.3) | 52(14.3) | 21(36.2) | |
| VHT | 361(85.6) | 310(85.2) | 51(87.9) | |
| Radio announcement | 126(29.9) | 100(27.5) | 26(44.8) | |
| Others | 45(10.7) | 40(11.0) | 5(8.6) | |
| Aware of health education in the community (Yes) | 331(78.4) | 288(79.1) | 43(74.1) | 0.391 |
| Health programs carried out in the community | | | | 0.005 |
| Aware of Malaria programs in the community | 293(88.5) | 255(88.5) | 38(88.4) | |
| Aware of River blindness programs in the community | 18(5) | 10(3.5) | 8(18.6) | |
| Aware of Immunization programs in the community | 301(90.9) | 259(89.9) | 42(97.7) | |
| Aware of Family planning programs in the community | 251(75.8) | 215(74.7) | 36(83.7) | |
| Aware of Other programs in the community | 34(10.3) | 29(10.1) | 5(11.6) | |
| Mode of delivery Health program | | | | 0.001 |
| Community routine health dialogues | 73(22.1) | 59(20.5) | 14(32.6) | |
| Radio | 112(33.8) | 91(31.6) | 21(48.8) | |
| Community home visits | 307(92.8) | 268(93.1) | 39(90.7) | |
| Others | 4(1.2) | 3(1.0) | 1(2.3) | |
| Reasons for not participating in health education | | | | 0.002 |
| No financial motivation | 14(16.7) | 7(10.0) | 7(50.0) | |
| Time | 25(29.8) | 21(30.0) | 4(28.6) | |
| Lost interest | 38(45.2) | 35(50.0) | 3(21.4) | |
| No health risk. | 7(8.3) | 7(10.0) | 0(0.0) | |

P-values based on the Fisher Exact test.

### Factors associated with Community involvement in PES activities in Bududa District

The multivariable analysis result showed that maintaining district support (financial and supervision) for onchocerciasis community activities after eliminating the disease was associated with a 14 times higher likelihood of community participation in post-elimination surveillance activities (Adjusted Odds Ratio [AOR] = 14, 95% Confidence Interval [CI] = 2.5–81.7). Individuals who observed black flies recently had 8 times higher odds of participating in PES activities compared to those who saw them in previous years (AOR = 8, 95% CI = 1.5–42.5, $P$ = 0.014). Moreover, participants who reported seeing black flies approximately one month ago had a 3.8 times higher likelihood of being involved in PES activities (AOR = 3.8, 95% CI = 1.1–13.2). Residing in the area before 2016 demonstrated a significant association with community involvement in PES activities. Community drug distributors (CDDs) had 3.2 times higher odds of engaging in PES activities (AOR = 3.2, 95% CI = 1.01–10.1), while community volunteers had a 4.3 times higher likelihood of being involved in PES activities (AOR = 4.3, 95% CI = 1.03–17.9) as shown in Table 6.

**Table 6. Factors associated with Community involvement in PES activities in Bududa District.**

| Variable | COR (95%CI) | P-value | AOR (95%CI) | P-value |
|---|---|---|---|---|
| Considers Black flies to be a threat to the community (ref = No) | 3.0(1.7, 5.5) | 0.001 | 1.7(0.7, 4.2) | 0.221 |
| Aware of District support towards the onchocerciasis community activities (ref = No) | 12.5(4.2, 37.6) | 0.001 | 14.3(2.5, 81.7) | 0.003 |
| Knows what causes river blindness(ref = No) | 2.9(1.3, 6.3) | 0.007 | 1.8(0.6, 5.4) | 0.318 |
| Ever seen the Black flies | | | | |
| Years back | 1 | | 1 | |
| Months back | 3.1(1.2, 8.3) | 0.024 | 3.8(1.1, 13.2) | 0.034 |
| Days back | 8.5(2.8, 25.8) | 0.001 | 8.0(1.5, 42.5) | 0.014 |
| Have you ever experienced any of the onchocerciasis symptoms(ref = No) | 2.8(1.2, 6.6) | 0.017 | 0.5(0.1, 2.5) | 0.414 |
| Participated as Sub- County onchocerciasis focal person during the most recent intervention phase(ref = No) | 10.7(1.4, 82.7) | 0.023 | 0.2(0.01, 3.8) | 0.260 |
| Participated as Community supervisor during the intervention phase(ref = No) | 12.8(3.4, 48.4) | 0.001 | 5.3(0.8, 34.6) | 0.084 |
| Participated as Drug distributors during the intervention phase(ref = No) | 6.8(3.5, 13.0) | 0.001 | 3.2(1.01, 10.1) | 0.050 |
| Participated as Community volunteer during intervention phase(ref = No) | 15.4(4.8, 49.1) | 0.001 | 4.3(1.03, 17.9) | 0.046 |
| Never Participated in any activity during intervention phase(ref = No) | 0.2(0.1, 0.4) | 0.001 | 0.5(0.2, 1.4) | 0.165 |

(Odds ratios and p-values based on logistic regression)

From the data collected through FGD and KIIs, theoretical saturation was attained with 2 FGDs and 8 KII. Each FGD consisted of 8 participants, with ages ranging between 38–78 years, while that of KIIs ranged between 59 to 67 years. Among the male FGD participants, the median age was 51.5 years, while for females it was 49.5 years. The median age for the key informants was 58 years.

## Sustainability plan after the program ends

Participants indicated that after post-treatment surveillance activities, the program ended abruptly without proper communication on what was to be done during the post elimination surveillance period. This abrupt end of the program greatly affected PES activities because the key community stakeholders, such as community members, community supervisors, parish supervisors, and health workers, did not know what to do during the post-elimination activities as they were not guided and empowered. This was noted by **KII06 and (FGD1 M_R05)**

> . . . *"When this program began, they did not care about its sustainability, and when they stopped, they never came back to tell us as the health workers and the communities that the disease has been eliminated"* [**KII06**]

> . . . *"We reached a time when we failed to know what exactly we were supposed to do during the post-elimination surveillance period. We were not properly briefed on what activities we were supposed to carry out to ensure that this disease of onchocerciasis does not reappear in our communities. We had no idea of what to do next".* [***FGD1 M_R05***]

## Funding/Financial support is totally cut off

The participants highlighted the absence of adequate funding as a significant barrier that impeded the participation of the community in post-elimination surveillance (PES) activities. They expressed that the financial resources previously provided by the program were no longer accessible, thereby affecting the effective implementation of PES activities. Furthermore, community supervisors faced difficulties in conducting awareness campaigns and engaging in sensitization efforts due to insufficient financial support. These concerns were underscored by key informants KII05, KII08_HW, and FGD1_R07.

> . . . *"Since onchocerciasis was eliminated, even the little financial support that the funder used to provide to carry out activities like community sensitization stopped"* [**KII05**]

> . . . *"The challenge of being a health worker is that I cannot tell these people that go and talk about onchocerciasis . . . they will ask me, are you going to pay us? Facilitation gives someone the strength to go and work. Also, while in the community, people will need something"* [**KII08**]

> . . . *"Whenever we had community meetings, we used to buy some for the community members some soda, but now we invite them without anything. It's not easy, and the community members may not turn up unless there is a serious problem that calls for community help. Even as the VHT without financial support, it becomes hard for us to carry out PES community activities"* [***FGD1M_R01***].

It was further reported that some of the community and parish supervisors resorted to looking for work that could make them earn an income and enable them to provide their families with basic needs. This was said by KII05 and FGD1_R02.

*. . . "We stopped continuing with the activities because there was nothing completely to moti-vate us. We go out to work so that we can get money to take care of our family. I would not continue with the activity when I am not being given anything. How would I feed my family, if I spent all my time on such activities that are not paying, How will my family be main-tained?* [**KII05**]

*. . . "I can't go to the field for about 6 hours to implement the post-elimination activities with-out allowance, and yet I also need to work very hard to make sure that my children have what to eat. I can assure you that I* couldn't *go to the community and spend so many hours for no pay"* [**FGD1_R02**]**.**

## Accessibility

The participants further highlighted that certain communities selected for the implementation of post-elimination surveillance activities were characterised by challenging topography and steep and hilly terrain. This geographical feature posed significant difficulties for community supervisors in reaching out to these communities. The steep and hilly nature, combined with heavy rainfall, made certain communities inaccessible. Consequently, conducting post-elimi-nation surveillance activities in these communities became highly challenging, as the community volunteers lacked essential protective gear such as raincoats, gumboots, and umbrellas.

Moreover, some communities were situated at considerable distances from each other, pos-ing transportation challenges for parish supervisors who lacked means of transport. This was reported by KII02, a 59-year-old female, KII05, and FGD_F04.

*. . . "Sometimes there are hard-to-reach areas; I may find it difficult to climb these steep hills. So, I end up not going to communities that are on the steep slopes"* [**KII02**]

*. . . "Our community is hilly. It is very hard for one to move from one place to another, and it is worse during the rainy season for one to reach some places. One needs a rain court, umbrella, gumboots, but these things are not there"* [**FGD2F_R04**]

*. . . "Some of the supervisors lost morale because the places were far and hilly, and yet they were not given any means of transport"* [**KII02**]

*. . . "I would move long distances from one parish to another, trying to supervise the commu-nity supervisors to ensure that they are doing the post-elimination activities as required. But due to distance and hilly and steep terrain, it became difficult for me to go there"* [**KII05**]

## Poor referral mechanism of the suspected cases of onchocerciasis

The participants emphasized a significant challenge pertaining to the lack of appropriate refer-ral forms for suspected onchocerciasis patients, impeding their seamless transfer from the community to healthcare facilities. During the FGD 02 session involving participant F_R01, it was brought to light that VHTs resorted to utilizing referral forms intended for other pro-grams, such as the malaria program. Furthermore, KII 03 underscored the absence of feedback provided to VHTs following the reporting of individuals suspected to be affected by Onchocerciasis.

*. . . "and even the referral forms we have been not for onchocerciasis, we use those for malaria, and for new programs that come in like COVID-19, but the reality is that health workers do not know about onchocerciasis"* [**KII 06**]

*. . . "When the program ended, we were told they would give us the forms, but they did not give us. The ones we have are for VHTs for general referral to Bubungi health center for issues like malaria, wounds, and maternity"* [**FGD02F_R01**]

*. . . "Whenever I referred a patient who was suspected to be having onchocerciasis, the health workers were not giving us any feedback about the patient who was referred if the patient was really having onchocerciasis or not"* [**KII03**].

## No drugs and diagnostic equipment at the health facility

The participants conveyed that the absence of medication and diagnostic equipment for onchocerciasis at the healthcare facility posed a significant challenge. This limited the ability to confirm the presence of the disease in suspected patients through diagnostic testing. Moreover, participants highlighted the persistent unavailability of onchocerciasis medication at the healthcare facility since the conclusion of the program. These observations were reported by individuals identified as KII07, KII08, KII03, KII04, and FGD02 F_R01. Additionally, it was noted that. . .

*. . . "The suspected onchocerciasis patients referred did not get drugs in the health facility, because ever since onchocerciasis was eliminated all the drugs got finished"* [**KII03**]

*. . . "The community members keep asking me, now that onchocerciasis is gone, do you just talk about prevention, or is there medicine? Do not talk about onchocerciasis without bringing the medicine asserted"* [**KII04**]

*. . . "Onchocerciasis has a specific drug which they promised to keep at the health facility and would only be given after testing the suspected person, but unfortunately, the community supervisors could refer onchocerciasis suspected patient, but there was no drug, and the ministry at the health facility has never provided testing kits"* [**KII08**].

*. . . "We may suspect them, but how the health workers will test them for onchocerciasis if they have no kits. When onchocerciasis was eliminated, they told us that the medicine would be kept at the health facilities and the suspected cases were to be referred there.. . .but the drugs were not there. The health workers treat them for scabies or skin infections because they don't have the diagnostic equipment for onchocerciasis. They guess"* [**KII07**]

*. . . "If we suspect someone to be having onchocerciasis, we do not know where they can get the correct treatment because they need to be tested.* [**FGD1 M_05**]

## Motivation

The participants said that a notable factor impeding their ongoing engagement in onchocerciasis post-elimination surveillance was a lack of motivation. They emphasized that the absence of incentives posed a significant challenge to the successful implementation of post-elimination activities. These sentiments were further underscored by KII07, who emphasized that without any form of compensation, individuals refrained from continuing their involvement in post-elimination activities. Although the nature of the work was voluntary, the absence of motivational factors or tangible rewards acted as demotivating factors.

*. . . "I cannot lie to you that community drug distributors and the parish supervisors are still doing their voluntary work of ensuring that post-elimination activities are implemented, how they can do voluntary work when a child at home wants to eat, they need sugar, soap. And,*

*when they were leaving, they did not bid farewell to us by appreciating us, so if they didn't appreciate us, how can I now tell volunteers to continue with the Post elimination activities"* [**KII07**]

## Discussion

In our study, we found that community involvement in PES was low at 14%. The main factors associated with involvement were district support, seeing black flies in the environment in the previous week of the survey, the previous month of the survey, and being a community volunteer.

Numerous factors contributed to the low level of community involvement we identified in our study. While there is limited published data on this issue, especially compared to other Neglected Tropical Diseases (NTDs), such as guinea worm eradication efforts [13], our findings align with commonly reported challenges. These include insufficient funding, lack of motivation, inadequate program sustainability planning, and unavailability of essential drugs at health facilities. These factors significantly hinder community participation in post-elimination surveillance (PES). Surprisingly, despite a three-year Post Treatment Surveillance period involving education and awareness campaigns in Bududa district, focusing on monitoring disease recrudescence and the presence of disease vectors, community involvement remains remarkably low.

The community engagement was intended to continue throughout the post-elimination surveillance. The lack of community involvement in post-elimination surveillance activities can be attributed to a diminished sense of motivation resulting from the discontinuation of funding by the Carter Center. These findings have significant implications, as they suggest a heightened risk of disease resurgence and potential delays in the certification of Uganda. Previous successful elimination programs, such as the Guinea worm eradication campaign, have demonstrated the critical role of community engagement and involvement [13]

Despite the limited extent of community engagement documented in this study, the findings demonstrate a significant correlation between district support for community onchocerciasis activities and community participation in post-implantation surveillance (PES) activities. The involvement of district-level supervision serves as a drive for community members to engage actively in PES activities. This finding aligns with previous research conducted by Cao et al. (2021) during the malaria elimination efforts in China [14], suggesting that the success of disease elimination programs relies on the active involvement of governmental leaders who can identify and address technical and financial requirements.

Our study also reveals a significant association between the regency of blackfly sightings and participation in PES activities. Specifically, individuals who observed black flies more recently exhibited a higher likelihood of engaging in PES activities compared to those who reported earlier sightings. This intriguing finding can be attributed to the fear of being bitten by infected black flies and contracting the disease. Additionally, it may be attributable to the profound impact of extensive health education campaigns, which were raised among communities regarding the black fly and its associated dangers. This result is consistent with a Knowledge, Attitudes, and Practices (KAP) study conducted in Nigeria, where the researchers found a significant relationship between community knowledge about the black fly and their engagement in disease control efforts [15]

Individuals involved in community drug distribution demonstrated a 3.2 times higher likelihood of engaging in PES activities. This heightened involvement can be attributed to their voluntary spirit, which initially motivated them to offer free services to the program.

Additionally, many of them were appointed as Village Health Team (VHT) members following the conclusion of the program, potentially contributing to their sustained involvement in certain aspects of the onchocerciasis program. An alternative explanation could be their comprehensive understanding of the disease's progression and severity, which compelled them to continue their engagement in PES activities to prevent onchocerciasis resurgence.

Similarly, the study indicates that community volunteerism significantly correlates with participation in onchocerciasis post-implantation surveillance activities. Specifically, individuals who volunteered within the community were 4.3 times more likely to engage in PES activities. This can be attributed to their previous roles as community volunteers or health educators, which kept them closely connected to the program even after funding ceased. It can be inferred that prior knowledge and previous participation played a crucial role in motivating them to contribute actively to PES activities.

Our study revealed the absence of sustainability plans at the community level for the continuation of post-elimination surveillance (PES). Participants expressed that the abrupt termination of the program by the funders without clear communication on the actions to be taken during the PES phase led to significant challenges in implementing PES activities. This lack of sustainability planning and empowerment of key stakeholders during the post-elimination activities has had a detrimental impact on the program. It is important to emphasise the significance of sustainability in disease elimination programs, particularly when they rely on donor funding.

A major obstacle encountered during the onchocerciasis post-elimination surveillance phase was the lack of funding. Participants highlighted the considerable challenge posed by insufficient financial resources at this stage, noting that even the minimal funding previously allocated to onchocerciasis activities had ceased. Consequently, community supervisors and drug distributors were unable to engage in community sensitisation for post-elimination surveillance activities due to the absence of facilitation. Adequate financial support has consistently played a pivotal role in disease elimination programs, particularly during surveillance efforts. For example, the successful elimination of Guinea worm in Uganda relied heavily on political support and financial incentives for case reporting [13] and malaria in China [16].

Additionally, the study unveiled the absence of a functional referral feedback mechanism for suspected onchocerciasis cases in Bududa district. Participants reported the lack of referral forms specifically designed to refer suspected onchocerciasis cases from the community to health facilities for diagnosis and treatment. The existing referral forms were intended for other programs, such as malaria control. Moreover, participants observed that referring parties received no feedback regarding further management or findings. An efficient referral system is vital for the effective functioning of a healthcare system at the local, provincial, and national levels. In the context of disease elimination programs, proper referral and surveillance forms are essential for accurate reporting and case identification [17].

Furthermore, the research highlighted the inadequate availability of drugs (Ivermectin) and diagnostic equipment for testing suspected onchocerciasis cases referred to health facilities in Bududa district. Participants expressed that referred patients were often not tested or provided with medication due to this issue. The guidelines outlined by the Ministry of Health [18] stipulate that health facilities should be equipped with the necessary resources to treat suspected cases referred through the village-based surveillance system. However, the findings contradict Ministry of Health guidelines, as participants reported a lack of medicine at the health facilities. Consequently, without the confirmation of suspected cases through testing, the administration of appropriate treatment becomes challenging.

Lastly, a notable challenge hindering PES activity was the lack of motivation during the post-elimination surveillance phase. Participants reported that the absence of facilitation

during this period demotivated them from carrying out PES activities. Consequently, many participants turned to alternative income-generating activities to meet their families' basic needs. The lack of motivation, coupled with limited financial support resulting from no funding for PES, raises concerns about the potential stagnation of PES implementation and the regression of progress achieved over the years. This finding is in line with another study by Krentel and others [19], who found a noticeable decline in the motivation of some of the community volunteers, potentially negatively impacting the success of elimination programs.

It is crucial to recognize that motivation plays a significant role in engaging community health workers and enhancing their performance [20]. Moreover, as emphasized by Abraham Maslow [21], understanding the psychological processes involved in employee motivation is essential for effective management and directing individuals towards organizational goals. Consequently, the limited involvement in PES activities can be attributed to community members prioritizing activities that provide personal benefits.

## Strengths and limitations

This study marks the initial exploration into the feasibility of community involvement in onchocerciasis post-elimination activities in Uganda. The data gathered herein holds the potential to enhance and refine post-elimination endeavours across all onchocerciasis foci in Uganda and beyond, particularly in Africa, where the disease has been successfully eliminated. Employing a combination of qualitative and quantitative methods allowed for a comprehensive assessment of the feasibility of community involvement in Post-Elimination Surveillance (PES) activities, enriching the study's findings.

Despite these contributions, it is important to acknowledge certain limitations. The existing literature on studies related to disease post-elimination surveillance is relatively scant. The data collection tools utilized in this study were devised based on the researcher's knowledge, with some sections adapted from the health and demographic survey tool. The absence of a standardized tool may have impacted the study's ability to assess certain parameters comprehensively. Additionally, reliance on recall-based questions introduces potential limitations. Nevertheless, the inclusion of probing techniques aimed at delving deeper into the subjects addressed in the study helped mitigate these recall-related challenges.

## Conclusions

Onchocerciasis post elimination surveillance activities are not feasible without community involvement and financial motivation. Community involvement in PES by the community members was low in Bududa district. Our findings highlight the importance of district support, funding, motivation, and sensitization during Post-Elimination Surveillance as critical.

## Supporting information

**S1 Data. This is the dataset that was used to produce the results of this study conducted in Bududa district, Eastern Uganda.**
(XLSX)

## Acknowledgments

We want to acknowledge our dear research assistants for the great work they did during the data collection process. We also thank the study participants for allowing us to take part in this study and having willingly provided the information that was required to complete this study —the Carter Center for allowing me to pursue my studies.

## Author Contributions

**Conceptualization:** Annet Tabitha Khainza, David Soita, David Okia, Joseph KB Matovu, Yovani Lubaale, Benon Wanume, David Mukunya, Peter Olupot-Olupot.

**Data curation:** Annet Tabitha Khainza, David Okia, Benon Wanume, David Mukunya, Peter Olupot-Olupot.

**Formal analysis:** Francis Okello, Edson Byamukama, Ambrose Okibure, Jimmy Patrick Alunyo.

**Funding acquisition:** Annet Tabitha Khainza.

**Investigation:** Annet Tabitha Khainza, David Soita, Francis Okello, Joseph KB Matovu, Yovani Lubaale, Edson Byamukama, Ambrose Okibure, Jimmy Patrick Alunyo, David Ogutu, Peter Olupot-Olupot.

**Methodology:** Annet Tabitha Khainza, David Soita, David Okia, Francis Okello, Joseph KB Matovu, Yovani Lubaale, Edson Byamukama, Ambrose Okibure, Jimmy Patrick Alunyo, Benon Wanume, David Ogutu, David Mukunya, Peter Olupot-Olupot.

**Project administration:** Annet Tabitha Khainza, David Soita, David Okia, Joseph KB Matovu, Yovani Lubaale, Benon Wanume, David Mukunya, Peter Olupot-Olupot.

**Resources:** Annet Tabitha Khainza.

**Software:** Francis Okello, Ambrose Okibure, Jimmy Patrick Alunyo.

**Supervision:** David Soita, David Okia, Joseph KB Matovu, Benon Wanume, David Mukunya, Peter Olupot-Olupot.

**Validation:** Joseph KB Matovu, Yovani Lubaale, Benon Wanume, David Mukunya, Peter Olupot-Olupot.

**Visualization:** David Ogutu.

**Writing – original draft:** Annet Tabitha Khainza, Ritah Nantale.

**Writing – review & editing:** Annet Tabitha Khainza, David Soita, David Okia, Francis Okello, Joseph KB Matovu, Edson Byamukama, Ambrose Okibure, Jimmy Patrick Alunyo, Ritah Nantale, David Ogutu, David Mukunya, Peter Olupot-Olupot.

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
