## [Decision Letter · Decision Letter 0]

8 Feb 2024

Dear Ms Khainza,

Thank you very much for submitting your manuscript "Community Involvement in Post-elimination Surveillance in Bududa District, Eastern Uganda: a cross-sectional study" for consideration at PLOS Neglected Tropical Diseases. As with all papers reviewed by the journal, your manuscript was reviewed by members of the editorial board and by several independent reviewers. In light of the reviews (below this email), we would like to invite the resubmission of a significantly-revised version that takes into account the reviewers' comments. 

We cannot make any decision about publication until we have seen the revised manuscript and your response to the reviewers' comments. Your revised manuscript is also likely to be sent to reviewers for further evaluation.

Sincerely,

Matthew C Freeman, MPH, Ph.D.

Academic Editor

Cinzia Cantacessi

Section Editor

Reviewer's Responses to Questions

**Key Review Criteria Required for Acceptance?**

**Methods**

-Are the objectives of the study clearly articulated with a clear testable hypothesis stated?

-Is the study design appropriate to address the stated objectives?

-Is the population clearly described and appropriate for the hypothesis being tested?

-Is the sample size sufficient to ensure adequate power to address the hypothesis being tested?

-Were correct statistical analysis used to support conclusions?

-Are there concerns about ethical or regulatory requirements being met?

Reviewer #1: • The authors stated the objective of the study. In the later part of the introduction, the authors wrote that ‘it is uncertain on the role and extent of the communities in Post elimination surveillance (PES). Also, it is not known whether the capacity created during Post Treatment Surveillance (PTS) period is sufficient to enable the communities to implement sound and vibrant PES activities’. However, the authors did not mention the roles and functions of the communities in PES and the capacity created during PTS. I kindly suggest that the authors include this information to justify the reason for their study. 

• Authors have obtained permission from the appropriate ethical regulatory body for the study. Please let the ‘ethics approval and consent to participate’ statement appear at the methods section of the manuscript and not to the end. 

• The authors clearly described the appropriate population suitable for the study. 

• The sample size was determined from sample size calculation and is sufficient to ensure adequate power to address the hypothesis being tested.

- The first sentence that begin paragraph 2 under the background of page 1 is not clear and understandable. I suggest that the authors re-write it with clarity to the readers.

- The statement ‘were replicated in Africa’ that ends the sentence; Ivermectin treatment for at-risk populations,----- of page 1 is not clear.

- Kindly delete one of the ‘in’ in the sentence in page 1

- Please this sentence ‘At the commencement of Uganda’s elimination efforts in 2007, a population of 4.9 million people from 16 foci excluding Victoria Nile focus is approximately 80 km from the source of River Nile near Jinja town extending to a point where the river became sluggish before reaching Lake Kyoga ) faced the risk of onchocerciasis. However, as of August 2023, this number has dwindled to 697,032 people in the Country’ is too long and not clear. I suggest the authors break it into smaller sentences with clarity. The authors should also cite the source of the information.

- There should be a full stop sign before the beginning of this sentence: ‘By close of 2017, three countries -Republic of Venezuela, Uganda, and Sudan had already ceased mass drug administration (MDA) and completed three years of post-treatment surveillance in at least one transmission area’. I also suggest the authors cite the source of this information.

- Please the first appearance of these words; ‘PES’ and ‘PTS’ should be written in full and the abbreviated forms in bracket after which the abbreviated forms alone can subsequently be used in the write-up.

- Please this sentence: ‘As of 2023, the country still has 7 districts in PTS, while 5 border districts are under transmission interruption suspected (interventions still on-going)’ is not clear. I suggest the authors re-write it with clarity.

- ‘Onchocerciasis’ is presented in the manuscript either in/not in italic. I suggest it should not be italicized.

Reviewer #2: The description of the study settings in Bududa district is comprehensive and highlights the significant achievements in onchocerciasis prevention. To further enrich this section, it would be beneficial to include details about the pre-control level of endemicity (hypo-, meso-, or hyperendemicity) and the specific interventions or ecological changes that contributed to onchocerciasis elimination. Such information could greatly aid in understanding the context and implications for the success and interpretability of post-elimination surveillance (PES). I recognise there is already some, specific information regarding this in the sampling procedure for the two sub-counties sampled. 

The study population's description, including the exclusion criteria and sample size estimation, is well-articulated. The rationale behind selecting the two sub-counties for sampling, considering their proximity to blackfly breeding sites and high endemicity, is sound and justifiable.

In terms of statistical analysis, the approach seems adequately described and appears to align with the study's objectives. Nevertheless, for a clearer understanding of the metrics used in the study, elaborating on what constitutes a "yes" in terms of community involvement in post-elimination surveillance would be useful. Although this is addressed later in the results and discussion, understanding the basis of these criteria—whether they stem from an open question or a predefined set of activities—would provide greater clarity to the reader.

Ultimately, the mixed-methods study designs contributes to the understanding of the topic in question and the ethical approval seems adequate.

**Results**

-Does the analysis presented match the analysis plan?

-Are the results clearly and completely presented?

-Are the figures (Tables, Images) of sufficient quality for clarity?

Reviewer #1: • The results are not clear, the tables are not formatted well and cannot be understood. The authors also failed to indicate the statistical test they used to analyze their results.

• For instance, Table 3, 4, 5 contain variables that do not add up to the respective totals as indicated by the authors. The total percent computed is either less than or more than 100%

• Table 5 appeared before Table 4 and some of the tables are overlapping which is incorrect.

• I suggest that the test statistics employed in the analysis of the results should be spelt out clearly under ‘Data analysis’ statement. I also suggest the statistical test used for the analysis of the results should be provided under each table. 

• Please delete the statement ‘of data collection’ that appeared at the end of the sentence under ‘Study design’.

• The authors failed to provide citation to the statements made under ‘study setting’. I suggest that the authors cite any piece of information and ideas in the manuscript that are not theirs.

• The sentence ‘Most of the participants were the original inhabitants of the area reported at 66% as detailed in Table 2’ is not clear. Kindly present this with clarity.

Reviewer #2: I was unable to locate any reference in Table 2 to the mentioned variable stating '66% of the participants were original inhabitants of the area.'

he 'Knowledge and Awareness about Onchocerciasis Disease' section provides valuable insights into the local population's understanding of the disease and anecdotal evidence of the continued presence, albeit limited, of the 'Simulium' vector. This suggests a potential risk of the disease's resurgence. 

Additionally, the 'Knowledge on Implementation of Onchocerciasis PES Activities' section is informative. It may be worthwhile to explore in the discussion the potential of epilepsy prevalence as an indicator of resurgent onchocerciasis transmission in PES, especially given its high association with onchocerciasis in regions of Uganda, South Sudan, and Cameroon (among other countries). 

A particular sentence appears incomplete: 'The community was 14 times more likely to be engaged in PES activities [AOR = 14, 95% CI = 2.5-81.7].' Perhaps it could be concluded with a phrase like 'if aware of district support for onchocerciasis community activities.' 

The qualitative analysis effectively highlights challenges in sustaining PES post-PTS, notably funding constraints, referral mechanisms, and motivation issues, alongside the selection of accessible communities, though the focus here should ideally be on a representative community sample."

**Conclusions**

-Are the conclusions supported by the data presented?

-Are the limitations of analysis clearly described?

-Do the authors discuss how these data can be helpful to advance our understanding of the topic under study?

-Is public health relevance addressed?

Reviewer #1: • The authors provided conclusions which are supported by the data presented in the manuscript. The benefit of the findings to the advancement of knowledge on the topic under study have been discussed and public health relevance has also been touched on. However, the limitation of the study is not mentioned. I suggest that the authors provide information on the limitations of the study and this information should come before the conclusion statement.

Reviewer #2: In the initial discussion paragraph, the categorisation of individuals who observed blackflies in their vicinity as part of, or associated with, post-elimination surveillance (PES) warrants further discussion. How does this relate to the possibility of onchocerciasis ressurgence in the area? If correlated with the possibly permanent vector disappearance (or reduction) in parts of Uganda, then it would be beneficial to mention so. 

With the limited literature available on PES, the study benefits from comparing its findings with those of similar diseases targeted for elimination and critically interpretes it. It's also important to note that the Bududa community received education and awareness campaigns during PTS, a detail that could have been mentioned before, perhaps within the methodology section (when describing the community).

The correlation between seeing blackflies and increased engagement in PES activities is intriguing. It raises the question of whether this observation is more indicative of heightened awareness rather than actual blackfly prevalence. Additional information on the presence or reduction of blackflies in the district would offer valuable context to the reader.

**Editorial and Data Presentation Modifications?**

Reviewer #1: I suggest the authors take into consideration the recommendations made in the methods and results section to improve on the manuscript.

Reviewer #2: Title: I recommend including "onchocerciasis" or "Onchocerca volvulus" in the title to specify that the study focuses on the elimination of this specific parasite.

Abstract: The abstract is clearly structured, effectively outlining the study's introduction, aims, methodology, results, and conclusion. However, elaborating on the role and implications of community involvement in post-elimination surveillance in the background section would provide more depth. If space is a concern, consider shortening the first two lines of the abstract, which may be more appropriate for the introduction/background of the study.

Background: 

1) In the initial discussion of onchocerciasis clinical manifestations, it would be beneficial to include a mention of the strong association between onchocerciasis and neurohormonal conditions, specifically 'onchocerciasis-associated epilepsy.' This link has been explored in previous studies in Uganda and could beneficial for the current research.

2) Regarding the sentence on the elimination of the disease in the Americas: 'However, by 2012, through the implementation of bi-annual Ivermectin treatment for at-risk populations, the disease was successfully eliminated in the Americas,' it would be informative to clarify that ivermectin was administered biannually and in some cases quarterly in the Americas. This detail could help explain the success in eliminating onchocerciasis in that region, in contrast with the challenges faced in some parts of Africa (more endemicity, effective vectors, logistical challenges in increasing CDTI frequency, etc.).

3) Additionally, in the statement 'The African continent adopted a comparable strategy, using Ivermectin to treat affected communities, but complemented it with a vector control/elimination initiative,' it would be clearer to specify that only certain parts of Africa, particularly those within the Onchocerciasis Control Program in West Africa (OCP), implemented vector control measures.

4) The background section would also benefit from a detailed description of the journey from onchocerciasis control to elimination. This should include an explanation of the definitions of post-treatment surveillance (recommended for at least 3 years to acheive onchocerciasis elimination according to the 2016 WHO guideliens) and post-elimination surveillance, and how the latter is currently conducted. For further insights, I recommend reviewing the paper 'Onchocerciasis Elimination: Progress and Challenges' by Lakwo et al., especially figure 1.

5) Finally, while the abbreviation PES is fully explained in the abstract, it should also be explicitly defined as 'post-elimination surveillance' when first used in the main text (e.g. in 1 city have attained Onchocerciasis elimination status and are under PES [8].).

Overall remark: The manuscript is overall understandable but could benefit from a comprehensive review to further enhance its linguistic clarity. This should include correcting punctuation, unnecessary capitalisation and addressing spacing issues. For example, in the sentence 'cover the entire country before declaring it free of Onchoc

---

## [Decision Letter · Decision Letter 1]

27 Apr 2024

Dear Ms Khainza,

Thank you very much for submitting your manuscript "Community Involvement in Onchocerciasis Post-elimination Surveillance in Bududa District, Eastern Uganda: a cross-sectional study" for consideration at PLOS Neglected Tropical Diseases. As with all papers reviewed by the journal, your manuscript was reviewed by members of the editorial board and by several independent reviewers. The reviewers appreciated the attention to an important topic. Based on the reviews, we are likely to accept this manuscript for publication, providing that you modify the manuscript according to the review recommendations. 

We appreciate the work you have put in to revise the file, but please take specific note of the comments from reviewer 1 both from the original submission, which don't appear to have been adequately addressed. When addressing the comments in the response letter, it would be helpful for you to indicate the specific changes in the table, rather than merely referring us to the page number of the manuscript. In addition, line numbers would help with the review. Finally, the reviewer has suggested an external editor review to improve clarity.

Sincerely,

Matthew C Freeman, MPH, Ph.D.

Academic Editor

Cinzia Cantacessi

Section Editor

We appreciate the work you have put in to revise the file, but please take specific note of the comments from reviewer 1 both from the original submission, which don't appear to have been adequately addressed. When addressing the comments in the response letter, it would be helpful for you to indicate the specific changes in the table, rather than merely referring us to the page number of the manuscript. In addition, line numbers would help with the review. Finally, the reviewer has suggested an external editor review to improve clarity.

Reviewer's Responses to Questions

**Key Review Criteria Required for Acceptance?**

**Methods**

-Are the objectives of the study clearly articulated with a clear testable hypothesis stated?

-Is the study design appropriate to address the stated objectives?

-Is the population clearly described and appropriate for the hypothesis being tested?

-Is the sample size sufficient to ensure adequate power to address the hypothesis being tested?

-Were correct statistical analysis used to support conclusions?

-Are there concerns about ethical or regulatory requirements being met?

Reviewer #1: No comment

Reviewer #2: Study Design: Regarding "This was a cross sectional study that utilized both quantitative and qualitative data collection methods."perhaps a more in-depht information could be given by mentioning the semi-structured questionnaire for the quantitative part and focused group discussions/kei informant interviews for the qualitative part.

As mentioned before, the mixed-methods study design is beneficial. Moreover, the ethics approval and consent is now clearly described. The sample size is clearly described and produced, although with a typo. I believe the authors mean 50.0% instead of 0,50% of expected community involvement. As one would only need 20 participants for a proportion of 0.50%.

**Results**

-Does the analysis presented match the analysis plan?

-Are the results clearly and completely presented?

-Are the figures (Tables, Images) of sufficient quality for clarity?

Reviewer #1: • Some of the tables are overlapping which is incorrect. I suggest that if a table cannot fit into one page, the authors should break that into sections maintaining the table and columns headings.

• I suggest that the statistical tool/s employed in the analysis of the results should be spelt out clearly under ‘Data analysis’ statement. 

• The description of the results under Implementation of Onchocerciasis PES Activities’, is not found in the table. For instance, where can 181 (42.9%) be found in the table. Also, when describing /interpreting the results, provide reference to the Table or Figures where they can be found.

Reviewer #2: The authors now better described the regression section.

In the discussions, a more nuanced approach may be needed, particularly to better differentiate it from the results sections, including adding more references.

Referral for oncho still not practiced in this PES area, and this is indeed an important issue that needs to be addressed and prevented. It was mentioned that there were referred suspected oncho cases. Is it known about their whereabouts, that is, if the diagnosis was confirmed and, if so, treated? As mentioned, it is indeed important to have ivermectin available and the doctors aware of onchocerciasis and its clinical manifestations.

**Conclusions**

-Are the conclusions supported by the data presented?

-Are the limitations of analysis clearly described?

-Do the authors discuss how these data can be helpful to advance our understanding of the topic under study?

-Is public health relevance addressed?

Reviewer #1: No comment

Reviewer #2: Limitations of the study are now mentioned.

The conclusions are supported by the data presented in the manuscript and can benefit the knowledge on the topic of PES of onchocerciasis.

**Editorial and Data Presentation Modifications?**

Reviewer #1: I suggest the authors take into consideration the recommendations made to enhance the clarity of the manuscript to the reader.

Reviewer #2: Introduction:

1) Break down long paragraphs (e.g., starting with "In 2017, the Global Burden of Disease study estimated...").

2) Mention "once or twice a year" for Ivermectin use in Africa compared to the Americas.

3) Avoid portraying temephos as entirely "chemically safe." Acknowledge potential environmental concerns.

4) "The positive outcomes observed in these affected regions can be attributed to strong collaborations among diverse stakeholders, including international donors, the WHO, governments, NGOs and communities (3){Gebrezgabiher, 2019 #1092}." I would say "partially attributed", as in the Americas there were also other drivers of successful onchcoerciasis elimination, such as less competent vectors.

5) In "As of 2023, the country still has 7 districts in Post-Treatment Surveillance (PTS)", consider using "had" instead of "has" for districts in past PTS.

4) Specify annual or biannual reasons for Ivermectin distribution in Uganda.

Overall English Language: The article requires substantial improvement, particularly in the introduction and, to a lesser extent, in the methodology. Consider:

1) Avoiding capitalization of words like "onchocerciasis," "ivermectin,", "post-", "annual", "country", etc.

2) Writing numbers one to ten in full text (e.g., "one to ten").

3) Addressing inconsistencies in text size and spacing.

4) Correcting punctuation (e.g., lack or excess of commas).

Abstract: Define AOR before using the abbreviation and mention the methodology used to calculate AOR (e.g. " logistic regression to assess factors associated with involvement").

Other points:

1) Ensure proper punctuation (e.g., "e.g. parasitic worm Onchocerca volvulus (O. volvulus) (1) (2)T his").

2) Avoid paraphrasing established terms (e.g., "ailment" instead of "disease").

3) Standardize spacing throughout the manuscript.

4) 

5) 

6) In "fisher exact where " suggest to mention it as "Fisher exact test". Great that the logistic regression is now mentioned in the data analysis plan

7) Add "to" after "communities implement sound" in the methods section.

8) Define Mount Elgon at first mention (excluding abstract), instead of Mt.. Provide a reference for its baseline prevalence if available.

9) Explain DDT abbreviation on first mention in the text (excluding abstract).

10) In "Annual mass treatment with ivermectin changed to two times per year along with vector elimination in 2007", clarify if vector control strategies were implemented in 2007, or if vector elimination was achieved in 2007.

11) In "and had the highest prevalence rate (68.2% for Bushika and 56.3 % for Bukigai). " I would suggest mentioning "at baseline" or "before control interventions". Also, a reference would be beneficial.

12) Define FGDs and KIIs on first mention in the article (excluding abstract).

13) Improve spacing and punctuation in results sections. Such as "more than half 216/422(51 .2%) were females " perhaps as "more than halve (216/422, 51.2%) were females".

14) In the result, it could be beneficial to have a background on the balckfly control history, such as how much was the ABR (bites per person per year) before control, how much did it decrease during control, and did it recover afterwards and how? (if info available) to link with the results on sights of blackflies of the current studies.

15) In the results and discussion, What is it meant by "district support"? Is it financial, logistical, supervision...? Perhaps define it in the methods or results.

16) In the discussion: "environment in a week [AOR 8, 95% CI= (1.5, 42.5)], in one month" should perhaps be "environment in the previous week to the survey [AOR 8, 95% CI= (1.5, 42.5)], in the previous month to the survey" ?

**Summary and General Comments**

Reviewer #1: The manuscript still needs some further work:

1. The authors failed to provide line numbers as recommended making it difficult to provide review comments to specifics in the manuscript.

2. The manuscript contains incorrect sentence constructions. The authors failed to provide full-stop at the end of some sentences, no ‘space’ to separate some words and sentences in the manuscript. 

3. The authors have also failed to provide information on authors’ contributions as recommended. I suggest the authors provide this information after the acknowledgement statement.

4. Please for the use of abbreviations in the write-up, kindly identify the first appearance of such words in the manuscript, write them in full and the abbreviated forms in brackets after which the abbreviated forms alone can subsequently be used in the write-up. Please the authors are encouraged to correct the anomalies in the manuscript.

5. The in-text citation is incorrect. It is square bracket not parenthesis. I suggest the authors correct it. 

6. The bibliographies have not been edited to suit the journal’s style as recommended. The authors should kindly follow the journal’s submission guidelines to format each of their bibliographies according to source.

Reviewer #2: This study offers valuable insights into community involvement in PES for onchocerciasis elimination. The mixed-methods approach is a strength.

The low community involvement identified highlights a potential gap between program focus on elimination and neglecting elimination surveillance activities. Additionally, logistical challenges in reaching remote areas may need to be addressed.

The study's generalizability is limited by the cross-sectional design and single district focus.

The findings on factors associated with involvement (district support, black fly sightings, volunteer experience) and identified challenges (funding, motivation, program sustainability, drug availability) are significant for improving community involvement in PES programs.

The authors should strengthen the discussion section with more references to support their claims.

Consider including page and line numbers for future revisions (facilitates commenting).

Encouragingly, the authors have addressed most of the concerns raised in a previous review. With further revisions to strengthen the presentation, discussion section and overall language clarity, this study has the potential to be a valuable contribution to the literature on community involvement in onchocerciasis elimination programs.

PLOS authors have the option to publish the peer review history of their article (what does this mean?). If published, this will include your full peer review and any attached files.

Reviewer #1: No

Reviewer #2: Yes: Luís-Jorge Amaral

Figure Files:

Data Requirements:

Reproducibility:

References

Please review your reference list to ensure that it is c

---

## [Editor Report · Decision Letter 2]

4 Jun 2024

Dear Ms Khainza,

We are pleased to inform you that your manuscript 'Community Involvement in Onchocerciasis Post-elimination Surveillance in Bududa District, Eastern Uganda: a cross-sectional study' has been provisionally accepted for publication in PLOS Neglected Tropical Diseases.

Best regards,

Matthew C Freeman, MPH, Ph.D.

Academic Editor

Cinzia Cantacessi

Section Editor

---

## [Editor Report · Acceptance letter]

26 Jun 2024

Dear Ms Khainza,

We are delighted to inform you that your manuscript, "Community Involvement in Onchocerciasis Post-elimination Surveillance in Bududa District, Eastern Uganda: a cross-sectional study," has been formally accepted for publication in PLOS Neglected Tropical Diseases.

Best regards,

Shaden Kamhawi

co-Editor-in-Chief

Paul Brindley

co-Editor-in-Chief
